# Transcriptional and Chromatin Accessibility Profiling of Neural Stem Cells Differentiating into Astrocytes Reveal Dynamic Signatures Affected under Inflammatory Conditions

**DOI:** 10.3390/cells12060948

**Published:** 2023-03-21

**Authors:** Maria Angeliki S. Pavlou, Kartikeya Singh, Srikanth Ravichandran, Rashi Halder, Nathalie Nicot, Cindy Birck, Luc Grandbarbe, Antonio del Sol, Alessandro Michelucci

**Affiliations:** 1Department of Life Sciences and Medicine, University of Luxembourg, L-4365 Esch-sur-Alzette, Luxembourg; 2Neuro-Immunology Group, Department of Cancer Research, Luxembourg Institute of Health, L-1210 Luxembourg, Luxembourg; 3Computational Biology Group, Luxembourg Centre for Systems Biomedicine, University of Luxembourg, L-4365 Esch-sur-Alzette, Luxembourg; 4Scientific Central Services, Luxembourg Centre for Systems Biomedicine, University of Luxembourg, L-4365 Esch-sur-Alzette, Luxembourg; 5Translational Medicine Operations Hub, Luxembourg Institute of Health, L-3555 Dudelange, Luxembourg; 6LuxGen Genome Center, Luxembourg Institute of Health & Laboratoire National de Santé, L-3555 Dudelange, Luxembourg; 7Computational Biology Group, CIC bioGUNE-BRTA (Basque Research and Technology Alliance), 48160 Derio, Spain; 8IKERBASQUE, Basque Foundation for Science, 48009 Bilbao, Spain

**Keywords:** neural stem cell, astrocyte, tumor necrosis factor, transcriptomics, chromatin accessibility, transcription factors

## Abstract

Astrocytes arise from multipotent neural stem cells (NSCs) and represent the most abundant cell type of the central nervous system (CNS), playing key roles in the developing and adult brain. Since the differentiation of NSCs towards a gliogenic fate is a precisely timed and regulated process, its perturbation gives rise to dysfunctional astrocytic phenotypes. Inflammation, which often underlies neurological disorders, including neurodevelopmental disorders and brain tumors, disrupts the accurate developmental process of NSCs. However, the specific consequences of an inflammatory environment on the epigenetic and transcriptional programs underlying NSCs’ differentiation into astrocytes is unexplored. Here, we address this gap by profiling in mice glial precursors from neural tissue derived from early embryonic stages along their astrocytic differentiation trajectory in the presence or absence of tumor necrosis factor (TNF), a master pro-inflammatory cytokine. By using a combination of RNA- and ATAC-sequencing approaches, together with footprint and integrated gene regulatory network analyses, we here identify key differences during the differentiation of NSCs into astrocytes under physiological and inflammatory settings. In agreement with its role to turn cells resistant to inflammatory challenges, we detect *Nrf2* as a master transcription factor supporting the astrocytic differentiation under TNF exposure. Further, under these conditions, we unravel additional transcriptional regulatory hubs, including *Stat3*, *Smad3*, *Cebpb*, and *Nfkb2*, highlighting the interplay among pathways underlying physiological astrocytic developmental processes and those involved in inflammatory responses, resulting in discrete astrocytic phenotypes. Overall, our study reports key transcriptional and epigenetic changes leading to the identification of molecular regulators of astrocytic differentiation. Furthermore, our analyses provide a valuable resource for understanding inflammation-induced astrocytic phenotypes that might contribute to the development and progression of CNS disorders with an inflammatory component.

## 1. Introduction

In the developing brain, multipotent neural stem cells (NSCs) can give rise to neurons, oligodendrocytes, and astrocytes. The process of differentiation towards a neurogenic or gliogenic fate is finely timed and regulated through mammalian brain development. During the expansion phase, NSCs self-replicate via symmetric cell division to expand their own pool. Later on, during mid-gestation they undergo asymmetric cell division and receive cues to differentiate into neurons. Lastly, from late-gestation to perinatal periods they enter the gliogenic phase and differentiate into astrocytes and oligodendrocytes [1,2]. Importantly, undifferentiated NSCs still reside in two niches of the adult brain, namely the subventricular zone (SVZ) of the lateral ventricles and the subgranular zone (SGZ) of the dentate gyrus [3].

Astrocytes are active supporters of the neuronal network, providing neurons with nutrients and metabolites, modulating neuronal synaptic transmission, participating in the formation and functioning of the synapses as well as directly communicating with neurons [4]. Further, astrocytes support the immune system and contribute to the formation of the blood brain barrier. Additionally, they contribute to oligodendrocyte differentiation and maturation as well as support myelination [5]. Therefore, aberrant astrocytic phenotypes can give rise to neurological diseases, including neurodevelopmental disorders [6]. For example, dominant gain-of-function mutations in the gene coding for glial fibrillary acidic protein (GFAP), the hallmark intermediate filament in astrocytes, elicit the accumulation of cytoplasmic GFAP aggregates resulting in macrocephaly, white matter degeneration, and developmental delay in Alexander disease [7,8]. During inflammatory events in the CNS, several immune mediators, including cytokines and free radicals, affect intrinsic NSC properties and neurogenesis [9]. Injuries of the CNS largely elicit proliferation and migration of NSCs toward the injury site, suggesting the activation of a putative regenerative response. However, detrimental cues encountered within the inflammatory niche are responsible, at least partly, for an incomplete regeneration from NSCs [9]. Thus, in traumatic brain injuries, for example, attempts to promote a regenerative-favoring environment using endogenous or transplanted NSCs for reparative purposes represent critical challenges [10,11]. In a different context, perturbations of the fine-tuned process of NSCs’ differentiation under inflammatory conditions can also give rise to brain tumors. Indeed, inflammation is linked to gliomagenesis [12] and evidence indicates that NSCs can give rise to gliomas [13,14]. Hence, defective differentiation abilities of NSCs and aberrant neoplastic astrocytic phenotypes in gliomas are attributed, at least partly, to an inflammatory microenvironment.

Tumor necrosis factor (TNF) is a crucial inflammatory cytokine mainly produced by microglia during neuroinflammatory processes. TNF induces various signaling responses within the cells that eventually result in apoptosis or necrosis [15], cell survival, differentiation, and inflammation in cells expressing the corresponding receptors [16]. Notably, the crosstalk between interferons and TNF affects the reprogramming of the macrophage epigenome and their inflammatory responses [17]. In this context, the role of epigenetic processes in the establishment, maintenance, and differentiation of NSCs is emerging [18]. For example, deficiency of the chromatin remodeler gene *Cdh5* in NSCs leads to their improper activation altering cell fate decisions and favoring an astroglial over a neuronal identity [19]. 

Here, we sought to investigate the dynamics of the transcriptional and chromatin accessibility changes underlying NSCs differentiation into astrocytes under normal and inflammatory conditions. For this, we adopted an integrated genome-wide approach combining RNA-sequencing (RNA-seq) and assay for transposase-accessible chromatin followed by deep sequencing (ATAC-seq) [20] to investigate the astrocytic differentiation process from murine neurospheres. We uncovered that normal differentiated astrocytes display mature cell-specific characteristics, whereas differentiating cells exposed to TNF do not reach the same level of maturity. In line with these observations and in agreement with a more restricted lineage-specific transcriptional program, we identified an overall reduction of chromatin accessibility in differentiated astrocytes compared to multipotent undifferentiated NSCs. Notably, this restriction was less pronounced under inflammatory conditions. Lastly, we reconstructed gene regulatory networks to identify transcription factors involved in the regulation of differentiation under normal and inflammatory settings.

Overall, our findings shed light on the molecular mechanisms underlying NSC differentiation into astrocytes and the effect of an inflammatory environment during this developmental process, which might pave the way to novel therapeutic targets restoring appropriate astrocytic phenotypes in CNS disorders with an inflammatory component, including neurodevelopmental diseases and brain tumors.

## 2. Materials and Methods

### 2.1. Animals and Ethics

C57BL/6J mice, both wild type and transgenic mice expressing green fluorescent protein (GFP) under the control of glial fibrillary acidic protein (GFAP) promoter were housed in individually ventilated cages (IVC) in a conventional animal facility at the University of Luxembourg in accordance with the EU Directive 2010/63/EU and Commission recommendation 2007/526/EC. We kept mice in groups under a dark–light cycle with ad libitum access to water and food. All the animal work of the present study has been conducted and reported in accordance with the ARRIVE (Animal Research: Reporting of In Vivo Experiments) guidelines to improve the design, analysis, and reporting of research using animals, maximizing information published, and minimizing unnecessary studies. We conducted all animal procedures in accordance with the local Committee for Care and Use of Laboratory Animals.

### 2.2. Cell Culture

We isolated primary NSCs from the ventricular zone of single wild type mouse brains (C57BL/6J, Harlan, The Netherlands) at embryonic day 14 (E14) as described before [21,22]. Primary cultures of neurospheres (NSPs) were kept under proliferating conditions in neurobasal medium (DMEM F12; Lonza, Basel, Switzerland) supplemented with 1% B27 without vitamin A (Life Technologies, Carlsbad, CA, USA), penicillin (100 U/mL; Lonza), streptomycin (100 g/mL; Lonza), and 20 ng/mL EGF (Epidermal Growth Factor; Life Technologies). We differentiated NSPs into astrocytes on 6-well or 12-well plates coated with poly-L-ornithine and by exchanging the proliferation medium with DMEM containing 10% FBS (Fetal Bovine Serum; Gibco, Grand Island, NE, USA), penicillin (100 U/mL; Lonza), and streptomycin (100 g/mL; Lonza). We differentiated cells into astrocytes at different time points (24, 48, 72 h and 1 week) at 37 °C in 5% CO_2_/95% air atmosphere, and we kept them under normal conditions or treated with TNF (50 ng/mL; R&D Systems, Minneapolis, MN, USA).

### 2.3. Selection of Astrocytes Based on Antibody-Coated Beads

We used NSPs and differentiating NSPs at different time points (24, 48 and 72 h) to isolate astrocytes by magnetic separation. We enriched astrocytic populations by magnetic separation using ACSA1+ (GLAST) beads (MACS Miltenyi Biotec) according to the manufacturer’s recommendations.

### 2.4. Real-Time qPCR (RT-qPCR)

We kept murine NSPs under proliferating conditions or we differentiated them into astrocytes for 24, 48, and 72 h or 1 week with or without TNF treatment. We extracted total RNA by the innuPREP RNA Mini Kit (Westburg) according to manufacturer’s recommendations. Complementary DNA (cDNA) was synthetized by using the ImProm-II Reverse Transcription System (Promega). We conducted cDNA synthesis for 5 min at 25 °C, followed by 1 h at 42 °C and then 15 min at 70 °C. We measured gene expression levels using SYBR Green Supermix (Promega) following the manufacturer’s recommendations and by using CFX Connect Real-Time PCR Detection System (Bio-Rad). The expression levels of mRNA were expressed as 2^−ΔCt^ and normalized to beta actin levels [23]. Primer sequences were as follows: *Mki67* forward primer: 5′-TTCCTTCAGCAAGCCTGAG-3′; *Mki67* reverse primer: 5′-GTATTAGGAGGCAAGTT-3; *Ccnb1* forward primer: 5′-AGAGGTGGAACTTGCTGAGCCT-3′; *Ccnb1* reverse primer: 5′-GCACATCCAGATGTTTCCATCGG-3′; *Egr1* forward primer: 5′-AGCCGAGCGAACAACCCTAT-3′; *Egr1* reverse primer: 5′-TGTCAGAAAAGGACTCTGTGGTCA-3′; *Gfap* forward primer: 5′-GGTTGAATCGCTGGAGGAG-3′; *Gfap* reverse primer: 5′-CTGTGAGGTCTGGCTTGG-3′; *Gpld1* forward primer: 5′-ACCCTAACCCAAGTCCTGCT-3′; *Gpld1* reverse primer: 5′-CAGGTCAGTCAGGTGCAGAA-3′; *Nfkbia* forward primer: 5′-GCCAGTGTAGCAGTCTTGAC-3′; *Nfkbia* reverse primer: 5′-GCCAGGTAGCCGTGAGTAG-3′; *Actb* forward primer: 5′-AGGGAAATCGTGCGTGACATCAAAGAG-3′; *Actb* reverse primer: 5′-GGAGGAAGAGGATGCGGCAGTGG-3′.

### 2.5. Immunocytochemistry

NSPs and differentiating cells were cultivated on poly-L-ornithine-coated coverslips and fixed with 4% paraformaldehyde in phosphate-buffered saline (PBS), followed by a permeabilization with 0.05% Triton X-100 in PBS. We performed a blocking step in PBS containing 3% bovine serum albumin (BSA) at 20–24 °C for 1 h. We conducted immunostainings by incubating cells with primary antibodies diluted in blocking solution overnight at 4 °C, followed by incubation with the corresponding secondary antibodies diluted in the blocking solution for 1 h at room temperature (1:1000; Jackson ImmunoResearch, West Grove, PA, USA). We used the following antibodies: rabbit anti-CD44 IgG (1:100; Abcam, Waltham, MA, USA), rabbit anti-Kir4.1 IgG (1:100; Alomone Labs, Jerusalem, Israel), rabbit anti-GLT1 (SLC1A2) IgG (1:100; Abcam), rabbit anti-GLAST (SLC1A3) IgG (1:100; Abcam), mouse anti-MAP2 IgG (1:200; Chemicon, Rolling Meadows, IL, USA) and mouse anti-O4 IgM (1:100; Bio-Techne). For GFAP staining, we used a cyanine 3-conjugated mouse anti-GFAP IgG antibody (1:400; Sigma, St. Louis, MO, USA). Cells were then washed and mounted with DAPI-Fluoromount G (SouthernBiotech, Birmingham, AL, USA). Images were collected by confocal microscopy using a LSM 510 confocal microscope (Carl Zeiss Micro Imaging, Jena, Germany) and analyzed on Adobe Photoshop (San Jose, CA, USA) software.

### 2.6. Preparation of Mouse Forebrain Cell Suspensions from hGFAP::eGFP Transgenic Mice and Flow Cytometry Analyses

*hGFAP*::eGFP transgenic mice (*hGFAP*::eGFP mice (FVB/N- Tg(GFAPGFP)14Mes/J) were purchased from The Jackson Laboratory (Jax stock #003257; Bar Harbor, ME, USA) [24]. The genetic identification of transgenic mice was determined by the analysis of DNA extracted from tails of 3-week old mice. DNA was extracted from tail samples (<5 mm) using “DirectPCR Lysis Reagent” (Viagen Biotech, Los Angeles, CA, USA) containing freshly prepared 20 mg/mL of proteinase K (Invitrogen, Merelbeke, Belgium). After a lysis incubation step at 55 °C for 5 h under gentle shaking (550 rpm), proteinase K was inactivated by incubating lysates at 85 °C for 45 min. DNA analysis was performed by Polymerase Chain Reaction (PCR). The PCR reaction mixture, with a total volume of 25 mL, contained 1 mL of DNA lysate, 12.5 mL of a PCR master mix (50 U/mL Taq DNA polymerase, 400 μM of each dNTP, 3 mM MgCl_2_; Promega; Leiden, The Netherlands), and 11.5 mL of primers mix (Eurogentec, Seraing, Belgium). Primer sequences were as follows: GFP forward primer: 5′-AAGTTCATCTGCACCACCG-3′; GFP reverse primer: 5′-TCCTTGAAGAAGATGGTGCG-3′; internal positive control forward primer: 5′-CTAGGCCACAGAATTGAAAGATCT-3′; internal positive control reverse primer: 5′-GTAGGTGGAAATTCTAGCATCATCC-3′. Initial denaturation at 94 °C for 1 min was followed by 35 cycles of denaturation at 94 °C for 30 s, annealing at 60 °C for 1 min and extension at 72 °C for 1 min, with final extension at 72 °C for 2 min. PCR samples were analyzed on a 2% agarose gel (ThermoFisher Scientific, Merelbeke, Belgium), revealed on a ChemiDoc XRS+ System (Bio-Rad, Temse, Belgium) and visualized using ImageLab software (Bio-Rad).

We collected forebrains from *hGFAP*::eGFP transgenic mice in calcium/magnesium-free HBSS at different postnatal developmental days (P4, P10, or P21). Tissue was diced and papain digested at 33 °C for 90 min (20 U/mL, Sigma) in dissociation buffer (EBSS (Sigma), D(+)-glucose 22.5 mM, NaHCO_3_ 26 mM and DNaseI 125 U/mL with EDTA 0.5 mM and L-cysteine–HCl 1 mM (Sigma)) and washed 3 times in dissociation buffer with BSA (1 mg/mL, Sigma) and trypsin inhibitor (1 mg/mL, Sigma) before mechanical dissociation through fire-polished Pasteur pipettes to a single cell suspension. Cells were pelleted, re-suspended in cold PBS with DNaseI at 1 × 10^6^ cell/mL, passed through a cell strainer (70-μm mesh; #352350, BD Falcon) and 7-aminoactinomycin D (7-AAD, Sigma) added. FITC-positive/PE-Cy5-negative cells were taken into account for quantification. We performed flow cytometry analyses using a FACS Aria I SORP running FACS Diva6.3 software (BD Biosciences, San Jose, CA, USA).

### 2.7. RNA-Sequencing Analyses

For RNA-sequencing analyses, murine NSPs were kept under proliferation conditions or were differentiated into astrocytes for 24 and 72 h without or with TNF treatment and for 1 week under normal conditions. We extracted total RNA with TRIzol (Invitrogen) following the manufacturer’s recommendations. We added a DNase treatment to the RNA extraction using DNA-free kit DNase Treatment and Removal Reagents (Ambion-Thermofisher cat #AM1906) following manufacturer’s instructions. We checked the RNA quality using a Fragment Analyzer System (Agilent Technologies, Santa Clara, CA, USA), and RQN were between 3.4 and 10. We converted total RNA into Stranded Total RNA Library for RNA-sequencing by using the TruSeq Stranded Total RNA Library Prep workflow with Ribo-Zero Gold kit according to Illumina’s instructions. Briefly, ribosomal RNA was removed from 300 ng of total RNA using biotinylated, target-specific oligos combined with Ribo-Zero Gold beads that deplete cytoplasmic and mitochondrial rRNA. Then, RNA was fragmented using divalent cations under elevated temperature of 94 °C. Cleaved RNA fragments are double-stranded cDNA converted, keeping strand-specificity. A single index adapter was ligated to each sample, and DNA fragments were enriched using a polymerase. All libraries have been quantified using a Qubit HS dsDNA kit (Thermofisher), and the library quality check has been performed using a High-Sensitivity NGS Fragment Analysis Kit on a Fragment Analyzer System (Agilent Technologies). We normalized indexed DNA libraries and pooled them at 10 nM. Libraries were paired-end sequenced at LuxGen Genome Center with 75 bp length on a NextSeq Illumina sequencer.

Fastq files contain raw sequenced data, and the FastQC tool was used for a quality check. The fastq data was trimmed to remove bad quality reads using Trim Galore and trimmed reads were then aligned to the transcriptome using STAR aligner. STAR produces unsorted BAM files as output, which were then sorted using Samtools. We used the sorted files as input for Feature Counts to obtain gene expression counts. We conducted differential gene expression analyses using DESeq2 package in R program (The R Foundation for Statistical Computing, version 4.0.4).

### 2.8. ATAC-Sequencing Analyses

Cells grown as either free-floating spheres (NSPs) or as astrocytes (for 24 h and 72 h non-treated or treated with TNF and for 1 week without TNF treatment) in poly-L-ornithine-coated plates were washed once with ice-cold PBS and were spun at 500× *g* for 5 min in a pre-chilled (4 °C) centrifuge. We then re-suspended the cells in 200 μL of lysis buffer (5 mM PIPES at pH 8, 85 mM KCl, 0.5% NP-40) and incubated them in a tube placed on ice for 20 min. Next, 800 μL of resuspension buffer (10 mM Tris pH 7.5, 10 mM NaCl, 3 mM MgCl_2_, 0.1% Tween-20) were added to the cells. Following cell counting, we centrifuged 50,000 cells at 500× *g* for 10 min in a pre-chilled (4 °C) centrifuge. The cell pellet was re-suspended in 50 μL of transposition mix (25 μL 2× TD buffer, 2.5 μL transposase, 17 μL PBS, 0.5 μL 10% Tween-20 and 5 μL water). Transposition reactions were incubated at 37 °C for 40 min in a thermomixer with shaking at 1000 rpm. Reactions were cleaned up with MinElute Reaction purification kit (Qiagen, Hilden, Germany) following manufacturer’s recommendations. Following purification, library fragments were amplified using 1× NEBnext PCR master mix and 0.6 μM of custom PCR primers 1 and 2, using the following PCR conditions: 72 °C for 5 min; 98 °C for 30 s; and thermocycling at 98 °C for 10 s, 63 °C for 30 s and 72 °C for 1 min. Full libraries were amplified for five cycles, after which an aliquot of the PCR reaction was used together with 9 μL of the PCR cocktail with Sybr Green at a final dilution of 0.05×. We ran this reaction for 30 cycles to determine the additional number of cycles needed for the remaining reaction. Libraries were purified using AMPure XP magnetic beads (Beckman Coulter, Brea, CA, USA), quantified using qubit fluorimeter and diluted to make a 4-nM final concentration. We sequenced pooled libraries on NextSeq500 75 bp paired-end reads. We used the FastQC tool to perform a quality check on fastq files containing raw sequenced reads, and bad-quality reads were removed using Trim Galore. The trimmed fastq reads were aligned to the genome using Bowtie2. We used Samtools to convert the SAM output files from Bowtie2 into sorted BAM files and to remove reads aligned to mitochondrial chromosomes. We used Model-based Analysis of ChIP-seq data (MACS) to call chromatin accessibility peaks, and the peaks that were found in the blacklisted regions of the genome (ENCODE) were removed before further analyses. Peaks were annotated using annotate peaks tool from HOMER and differential accessibility analysis was performed using DiffBind package in R program.

### 2.9. Transcription Factor Footprint Analysis

We used the Regulatory Genomic Toolbox to study the changes in the transcription factor binding motifs in open chromatin regions of the genome. Using the HINT tool [25], we identified transcription factor motifs in the epigenomic landscape. We conducted differential motif analysis using the rgt-motif analysis tool to identify transcription factor binding motifs differentially present across two conditions, thus providing information of underlying differences in gene regulation.

### 2.10. Reconstruction of Gene Regulatory Networks

We re-constructed gene regulatory networks and identified hubs based on topology analyses. First, we inferred gene regulatory networks (GRNs) at each time point by using the GRN inference tool GENIE3 [26], which utilizes random forest-based ensemble methods to identify regulatory links between genes. We performed bootstrapping on these interactions and only selected interactions with significant value of more than zero in more than 50 percent of runs. To obtain high confidence networks, we further overlaid the re-constructed networks from GENIE3 with manually curated interactions from the Metacore database. We further pruned the GRN by contextualizing it using a tool developed in our group: https://gitlab.lcsb.uni.lu/andras.hartmann/GRNOptR (accessed on 15 November 2021). Briefly, the tool utilizes a parent network and differential gene expression data with respect to the preceding time-point to create contextualized gene expression networks. Contextualization removes nodes whose expression cannot be justified by the nodes regulating it as input. We then identified the regulatory hubs by performing topological analyses using the igraph package in R. We further used the clusterprofiler package in R to perform Gene Ontology pathways enrichment of transcription factors included in the re-constructed GRNs. We visualized the resultant networks using Cytoscape.

### 2.11. Raw Data Files

We deposited ATAC-seq and RNA-seq datasets in the Gene Expression Omnibus (GEO) database with accession number GSE225729.

### 2.12. Statistical Analyses

Data were analyzed using GraphPad Prism 8 software (GraphPad software, La Jolla, CA, USA) and R environment (R Core Team, Vienna, Austria). The number of replicates required for comparison was estimated using power analysis (set 80% and *α* = 0.05) and guided by our previously published work [21]. We assessed normality using Kolmogorov–Smirnov and Shapiro–Wilk tests for normality. Depending on the number of averaged samples, the Central Limit Theorem was applicable, and even if data were not normal, the mean uncertainty was normally distributed. Unless otherwise indicated, all data are presented as mean ± standard deviation (SD) or standard error of the mean (SEM) of at least three independent biological experiments. We performed statistical analyses using unpaired *t* test or two-way ANOVA. All differences were considered significantly different at *p* value < 0.05 and were annotated as follows: * *p* < 0.05, ** *p* < 0.01.

## 3. Results

### 3.1. Characterization of an In Vitro Model of Astrocytic Differentiation from Murine Neurospheres

To investigate the molecular mechanisms underlying NSCs differentiation into astrocytes under inflammatory conditions, we first characterized our in vitro cellular model of primary neurospheres (NSPs) derived from NSCs isolated from the ventricular zone at embryonic day-14 (E14) of the mouse embryos. Briefly, we differentiated NSPs into astrocytes in the presence of 10% fetal calf serum (FCS) for one or two weeks and analyzed them at specific intermediate stages, including 24, 48, and 72 h (Figure 1A). Under these conditions, differentiating NSPs showed progressively increased expression levels of the astrocytic marker GFAP and a concomitant decreased expression of the stem cell marker CD44 (Figure 1B). After 2 weeks of differentiation, mature astrocytic markers, such as Kir4.1 (*Kcnj10*), GLT1 (*Slc1a2*), and GLAST (*Slc1a3*) were highly expressed (Appendix A). Notably, we did not detect neurons and oligodendrocytes in these culture conditions (Appendix A), while multipotentiality of the NSCs was confirmed by the appearance of these cells upon addition of 1% B27 and 2 ng/mL EGF to 0.5% FCS medium (Appendix A). At the transcriptional level, differentiating astrocytes at all stages showed significantly decreased expression levels of the proliferating (*Mki67* and *Ccnb1*) and stem cell (*Egr1*) markers compared to NSPs (Figure 1C). On the other hand, the expression levels of *Gfap* and the astrocyte-enriched gene *Gpld1* were mainly up-regulated in differentiating cells at the different time points. We detected high *Gfap* expression levels at 48 and 72 h, which dropped at 1 week, while *Gpld1* expression was not detectable in NSPs and showed progressively increased levels up to 1 week of differentiation (Figure 1D). In order to corroborate these results in vivo, we investigated the decrease of *Gfap* expression during astrocytic maturation by isolating GFP+ cells from *hGFAP::*eGFP transgenic mice. For this, we separated GFP+ cells by FACS at postnatal day 4 (P4), P10, and P21, with P4 representing an immature phase, P10 signifying an intermediate stage, while P21 corresponded to a mature astrocytic stage. In line with the in vitro model, the amount of *hGFAP::*eGFP+ cells gradually decreased along these time points (P4: 39.2 ± 3.6%; P10: 21.0 ± 0.9%; P21: 3.5 ± 1.3%) (Figure 1E). Notably, we confirmed the upregulation of *Gpld1* as a marker of astrocytic maturation in vivo from our previous transcriptomics analyses of GFP+ cells sorted from *Aldh1l1*::eGFP BAC transgenic mice when comparing cells isolated at P4, P10, and P21 (Appendix A) [27]. We similarly verified their progressive exit from the cell cycle by detecting the decrease of *Ccnb1* expression levels (Appendix A). Lastly, to verify the purity of our in vitro cultures, we sorted enriched astrocytic populations by MACS using an antibody against the pan-astrocytic marker GLAST (*Slc1a3*). GLAST+ cells exhibited similar expression levels of *Mki67*, *Ccnb1*, *Gfap,* and *Gpld1* to unsorted cells (Appendix A). Hence, this common gene expression pattern observed in GLAST+ and GLAST- cells suggests that our culture conditions give rise to pure astrocytic populations expressing GLAST levels consistent with their maturation state.

Taken together, these results demonstrate that our in vitro model of NSPs differentiation into astrocytes represent a reproducible method enabling enrichment of astrocytic populations expressing markers of ex vivo-isolated astrocytes.

### 3.2. Exposure of NSPs to TNF under Differentiating Conditions Modulates the Expression of Specific NSC and Astrocytic Markers

Next, we took advantage of our model to investigate the effect of inflammation during the differentiation of NSPs into astrocytes. For this, we cultivated the NSPs in differentiation medium with or without TNF (50 ng/mL), as in our previous study [21]. As expected, TNF treatment induced NF-κB activation in differentiating NSPs as shown by enhanced *Nfkbia* mRNA levels compared to untreated cells at the different time points (Figure 2A). While we detected differences in the expression levels of the stem cell marker *Egr1* in the presence of TNF when compared to untreated samples at 24 h and 1 week (Figure 2B), TNF treatment did not cause significant changes in the expression levels of *Ccnb1* (Figure 2C) and *Mki67* (Figure 2D). Further, *Gfap* and *Gpld1* were down-regulated in TNF-treated cells at 48 and 72 h of differentiation for the former, and at 72 h and 1 week for the latter (Figure 2E,F).

These targeted gene expression studies, together with our recent analyses [21], show that exposure of differentiating NSPs to TNF induces the activation of the NF-κB pathway without affecting the exit of the differentiating astrocytes from the cell cycle. On the other hand, TNF treatment causes transcriptional changes in the expression levels of specific stem cell and astrocytic markers, thus affecting the physiologic astrocytic differentiation process.

### 3.3. Transcriptomics Analyses under Normal and Inflammatory Conditions Reveal Discrete Populations of Differentiating Astrocytes

As TNF affected the expression of specific NSC and astrocytic genes during differentiation of NSPs into astrocytes, we sought to expand our analyses by conducting a genome-wide RNA-sequencing analysis of the astrocytic transcriptome at different time points, both under normal and inflammatory conditions. Principal component analysis (PCA) using the top 1000 variable genes showed distinct clustering corresponding to the different conditions (Figure 3A). We identified differentially expressed genes (*p* < 0.05) between different conditions and compared them at 24 and 72 h under normal or inflammatory conditions. Specifically, we detected 4187 overexpressed genes and 4544 down-regulated genes at 24 h compared to undifferentiated NSPs, both with or without TNF. On the other hand, we detected 1005 and 756 genes, respectively, exclusively up- or down-regulated at 24 h under control conditions, while 1227 and 922 genes were correspondingly enhanced or decreased only under TNF treatments. We obtained similar results at 72 h (Figure 3B). Hierarchical gene clustering of the differentially expressed genes during normal differentiation (up- and down-regulated genes, adjusted *p* value < 0.05) revealed five major clusters differentially represented across the specific time points (Figure 3C). Corresponding Gene Ontology (GO) enrichment analyses indicated that proliferating NSPs were mainly associated with cluster 1, largely characterized by terms related to RNA processing. Clusters 2 and 4, largely represented in NSPs and 24 h-differentiated astrocytes, were characterized by processes associated with regulation of cell morphogenesis, motility, and migration. Clusters 3 and 5, mostly enriched in differentiating astrocytes at 72 h and 1 week, were characterized by terms linked to cell projection assembly and cilium organization (Figure 3D).

We conducted similar analyses to investigate the effect of TNF treatment at 24 and 72 h. Hierarchical gene clustering of the differentially expressed genes during differentiation under normal and inflammatory conditions (up- and down-regulated genes, adjusted *p* value < 0.05) identified four major clusters differentially represented across the specific settings (Figure 3E). Clusters 1 and 4, characterized by terms associated with response to molecules of bacterial origin and regulation of inflammatory response, were enriched under TNF treatment, at both 24 and 72 h. On the contrary, cluster 2 described by terms linked to regulation of cell motility and migration, was under-represented under inflammatory conditions. Notably, we observed that the presence of TNF, especially when comparing conditions at 24 h, decreased cluster 3, related to gliogenesis, glial cell differentiation and development (Figure 3F).

Next, we used heat maps to visualize gene expression levels across different time points between normal and inflammatory conditions of specific genes associated with different astrocytic categories. Specifically, we analyzed markers linked to “developing and differentiated astrocytes”, “reactive astrocytes”, “A1-neurotoxic-astrocytes”, and “A2-neuroprotective-astrocytes” taking advantage of different available datasets [28,29,30,31] (Appendix A). Notably, astrocytic gene markers, including *Gfap*, *Slc14a1*, and *Gja1* were up-regulated at 72 h in the absence of TNF (Appendix A), while reactive astrocyte marker genes, including *Timp1*, *Lcn2*, *Icam1,* and *Ptx3,* were enhanced under inflammatory conditions, mainly at 24 h or *Sbno2*, *Cxcl10,* and *Cd109* [32,33,34] at 72 h (Appendix A). Further, we showed that the obtained astrocytic populations express a mixture of A1 and A2 markers, both under normal and inflammatory conditions, although the specific markers differ between the two conditions (Appendix A). Lastly, due to the importance of glycogen and glucose metabolism in astrocytes, we also looked at the expression levels of genes associated with “glycogen activity” and “glucose metabolism” (Appendix A). Along the differentiation process, we detected variable expression levels of genes related to glycogen activity (*Ugp2*, *Pygb*, *Ppp1r3c*) and glucose metabolism (*Lmbrd1*, *Pid1*, *Esr1*, *Upk3b*, *Lep)*, which were modulated under TNF exposure (Appendix A).

Taken together, these results show that genes linked to gliogenesis and glial cell development are down-regulated in the presence of TNF, suggesting that an inflammatory environment alters the physiological astrocytic differentiation, thus affecting critical associated functional activities, including glycogen and glucose metabolic pathways.

### 3.4. Chromatin Accessibility Profiling Detects Extensive Chromatin Remodeling along the Astrocytic Differentiation Affected by TNF Exposure

To understand further the mechanisms underlying changes in gene expression under normal and inflammatory conditions, we sought to examine the corresponding chromatin accessibility profiles, which affect, at least partly, gene expression depending on their more open or closed states at specific loci. To do this, we performed ATAC-sequencing analyses under the same conditions as the transcriptome studies. A first PCA analysis using differential peaks discriminated the various time points and highlighted differences between treated and untreated samples. Of note, similarly to the PCA analyses of the RNA-seq samples, PC1 seemed to separate samples according to the developmental stage, while PC2 divided them according to stage and treatment (Figure 4A). Next, we conducted our analyses based on differentially accessible peaks in the promoter region nearest to the transcription start site. We identified differentially accessible peaks (*p* < 0.05) up- and down-regulated between different conditions and compared them at 24 and 72 h under normal or inflammatory conditions. Specifically, we detected 545 up-regulated and 9081 down-regulated accessible peaks at 24 h compared to undifferentiated NSPs, both with and without TNF. On the other hand, we detected 102 and 960 accessible peaks, respectively, exclusively up- or down-regulated at 24 h under control conditions, while 297 and 385 accessible peaks were correspondingly enhanced or decreased only under TNF treatments. We found similar proportions at 72 h (Figure 4B). Hierarchical gene clustering of the differentially accessible peaks across the analyzed normal differentiation time points (up- and down-accessible peaks, adjusted *p* value < 0.05) revealed five major clusters differentially represented across the specific time points (Figure 4C). Clusters 1, 3, and 5, mainly associated with proliferating NSPs, were the highest represented in the heatmap, thus indicating that multipotent NSPs have higher chromatin accessibility when compared to differentiated astrocytes (Figure 4C). NSPs were characterized by GO terms related to RNA processing, histone, and covalent chromatin modifications (clusters 1 and 3), synapse organization and axonogenesis (clusters 3 and 5) as well as lipid and monovalent inorganic cation transport (cluster 5). On the other hand, cluster 2, mainly enriched in NSPs and 1 week-differentiated astrocytes, was represented by GO terms related to response to chemical stress and external stimulus. Lastly, cluster 4, mostly enhanced in late-stage astrocytes, showed terms linked to lipid localization and transport (Figure 4D).

Under inflammatory conditions, we detected increased chromatin accessibility in TNF-treated cells when compared to untreated cells. More precisely, following TNF treatment, prevalent clusters 1, 2, and 3 were accessible at both 24 and 72 h (Figure 4E). Enrichment analysis of these clusters specifically showed involvement of the NF-κB signaling pathway (cluster 2), results that are in agreement with the RNA-seq data. Cluster 4, which was under-represented under inflammatory conditions, was associated with GO terms related to cell junction assembly, while cluster 5, mainly enriched in 24 h-differentiated astrocytes, was linked to regulation of developmental growth (Figure 4F). Lastly, we used volcano plots to analyze differentially accessible peaks across time points between normal and inflammatory conditions. Here, a comparison between TNF-treated and -untreated samples showed a skewed distribution towards peaks that were more open under inflammatory conditions, both at 24 and 72 h (Appendix A).

Overall, we observed significant changes in chromatin accessibility upon astrocytic differentiation compared to undifferentiated cells, with genes associated, for example, with lipid localization and transport displaying increased accessible peaks in their promoter region. Upon TNF treatment, we detected a strong shift towards enhanced chromatin accessibility compared to normal conditions, with several gene clusters related to inflammation.

### 3.5. Transcription Factor Binding Analyses Uncover Dynamic Footprints Associated with Astrocyte Specification and Maturation under Normal and Inflammatory Conditions

Next, we investigated changes in the relative numbers of a given motif across accessible chromatin regions by footprint analyses using HINT-ATAC [25] to detect possible transcription factors (TFs) associated with astrocyte specification and maturation with or without TNF treatment.

We first analyzed differences in the number of TF binding motifs between NSPs and differentiating astrocytes at 24 and 72 h under normal conditions. At 24 h, we detected an enriched activity score for *Atoh1*, *Dlx1*, *Rarg,* and *Arid3a* with a concomitant decreased score for *Dmbx1* and *Tcf21* TFs when compared to NSPs (Figure 5A). Similar analyses conducted at 72 h showed increased activity score for *Hoxd8*, *Lhx3*, *Rarb*, *Rarg,* and *Dlx1*, together with a decreased score for *Hes1*, *Tcf21,* and *Tcfl5* TFs compared to NSPs (Figure 5B). Hence, based on these analyses, ATOH1 and ARID3A may represent TFs associated with astrocytic specification (24 h), HOXD8, LHX3, and RARB may represent TFs linked to astrocyte maturation (72 h), and DLX1 and RARG are likely TFs linked to both processes (Figure 5A,B).

By conducting similar analyses under inflammatory conditions, we detected at 24 h increased activity score for *Nfe2l2 (Nrf2)* and a decreased score for *Dmbx1*, *Arid3a*, *Arid3b,* and *Atoh1* TFs when comparing to normal conditions (Figure 5C). Lastly, at 72 h, *Atoh1* and *Rarg* TFs showed an enhanced score by comparing TNF-treated versus untreated cells, while the score for *Tcf21* and *Dmbx1* was decreased (Figure 5D). Due to the importance of *Nfe2l2 (Nrf2)* in regulating inflammatory responses, mainly by inducing the expression of antioxidants and cytoprotective genes, we analyzed its binding motifs across accessible chromatin regions at the genome-wide level. We detected enriched footprints across the whole genome in TNF-treated cells at 24 h (Figure 5E), thus suggesting that NRF2 may represent a master TF driving the astrocytic differentiation in the presence of TNF.

### 3.6. Transcriptional and Chromatin Accessibility States Positively Correlate at Specific NSC and Astrocytic Gene Loci

Next, we sought to study the correlation between transcriptional and chromatin accessibility states at the genome-wide level to explore to which extent the chromatin states are linked to the transcriptional activities at specific gene loci. For this, we took advantage of our RNA-seq and ATAC-seq data to conduct correlation analyses between gene expression levels and their chromatin accessibility states at 24 h. We used scatterplots to show the number of genes differentially expressed and exhibiting differentially accessible peaks at their promoter regions under normal (Figure 6A) and inflammatory (Figure 6B) conditions. By using Pearson’s correlation analysis for log-fold change in gene expression and accessibility, we identified similar correlation coefficients (normal: R = 0.38; inflammation: R = 0.51) to those detected in analogous studies comparing RNA-seq and ATAC-seq datasets (Ackermann et al., 2016, Starks et al., 2019). In a second step, we conducted a targeted correlation analysis at specific NSC, resting and reactive astrocytic gene loci [28,29,30,31]. Here, we calculated at 24 h a higher correlation score (R = 0.64) compared to genome-wide levels. Under normal conditions, we showed that specific NSC (e.g., *Ccnb1*, *Mki67*, *Egr1*) or astrocytic (e.g., *Gfap*, *Gpld1*, *Clcf1*) markers exhibited, respectively, decreased or increased gene expression levels and accessibility (Figure 6C). Under TNF exposure, we detected enhanced gene expression levels and accessibility for reactive astrocyte markers (e.g., *Lcn2*, *Cxcl10*, *Osmr*, *Sbno2*) (Figure 6D). Interestingly, *Nfkbia*, a pan-inflammatory marker, exhibited both enhanced gene expression levels and accessibility states (Figure 6D).

Taken together, these results indicate that at specific NSC and astrocytic gene loci, gene expression levels are associated with chromatin accessibility states, thus indicating a close link between chromatin remodeling and gene expression at lineage-specific gene loci.

### 3.7. Inference of Gene Regulatory Networks Enables the Identification of Key Transcription Factor Bindig Motifs Associated with Astrocytic Differentiation under Physiological and Inflammatory Conditions

To systematically identify regulators of astrocytic differentiation under physiological and inflammatory conditions, we reconstructed the corresponding gene regulatory network (GRN) for each analyzed time-point. The GRN after 24 h of physiological differentiation was represented by 262 interactions between 96 TFs, which, based on the topological analysis, include *Esr1*, *Nr3c1*, *Foxa1*, *Gata2*, *Stat5a*, *Fosl2*, *Stat1*, *Nfia,* and *Arid5b* (Figure 7A). Corresponding GO analyses highlighted terms associated to developmental processes, including “regulation of developmental process”, “tissue development”, and “regulation of cell differentiation” (Appendix A). At 72 h, when compared to 24 h, the GRN consisted of 91 interactions between 46 TFs, thus reflecting a lineage-committed network made up of less than half the TFs when compared to earlier stages (Figure 7B). The top regulatory hubs include *Rxra*, *Rarg*, *Cebpb*, *Myc*, *Smad3*, *Rbpj,* and *Esr1*. Subsequent GO analyses showed enriched terms related to cell responses and communication, such as “cellular response to stimulus”, “signaling”, and “signal transduction” (Appendix A), thus indicating that at earlier stages, differentiated cells went through the developmental process (astrocyte specification) and later acquired their functional astrocytic properties (astrocyte maturation). Interestingly, 10 TFs were shared between the two GRNs (Appendix A). Out of these, we identified *Esr1* and *Foxp1* as regulatory hubs at both time points.

Similarly, we reconstructed GRNs under inflammatory conditions. Following a 24-h differentiation, the GRN consists of 107 TFs and 393 interactions (Figure 8A). We identified *Stat3*, *Smad3*, *Nfkb1*, *Runx1*, *Ar*, *Cebpb*, *Relb*, *Nfkb2,* and *Atf3* as regulatory hubs. Out of these TFs, 75 of them, such as *Esr1*, *Nr3c1*, *Gata2*, *Stat5a*, *Fosl2*, *Sat1*, *Junb*, *Mef2c*, *Arid5b*, *Mef2a,* and *Foxo3*, were shared with the corresponding GRN under normal conditions (Appendix A), suggesting that the developmental process and the inflammatory response overlap during the acquisition of the astrocytic phenotype. GO analyses of the TFs underlying the GRN at 24 h under inflammatory conditions strengthen this observation, with terms associated to both developmental processes, such as “regulation of developmental process”, and inflammatory responses, including “immune system process”, being highly represented (Appendix A). Lastly, following a 72-h differentiation under TNF treatment, the GRN is composed of 18 interactions between 15 TFs, including *Rxra*, *Myc*, *Stat5b,* and *Rorc* (Figure 8B). Corresponding GO analyses identified “response to chemical”, “response to hormone” and “response to lipid” as main enriched terms (Appendix A). Among the resultant TFs, 12 of them, including *Rxra*, *Rarg*, *Stat5b,* and *Myc*, were in common with the corresponding network under physiological conditions, while 3 TFs (*Thrb*, *Zfp217*, *Zbtb37*) were unique to the inflammatory condition (Appendix A).

Taken together, reconstructed GRNs enabled the identification of regulatory hub TFs playing critical roles along the differentiation of astrocytes in the presence or absence of TNF. These analyses allowed establishing the effect of an inflammatory environment along the astrocytic development, indicating that it affects the underlying transcriptional signatures skewing astrocytes towards a discrete astrocytic phenotype when compared to physiological settings.

## 4. Discussion

Astrogliogenesis depends on both extracellular cues and intrinsic dynamic changes of the epigenome. Perturbations during astrocytic development may result in neurological diseases, including neurodevelopmental disorders, such as Rett and fragile X syndromes [35] or Down syndrome and autism spectrum disorders [36]. In recent years, computational approaches to study astrocytes and their development using next-generation sequencing technologies have enabled a better understanding of their phenotypic and functional role in the CNS [37,38]. Further, the generation of novel in vitro models, such as astrocytes derived from induced pluripotent stem cells, has opened up a new area for studying neurological diseases in vitro. However, as the brain hosts a range of astrocyte populations, it is critical to develop standardized protocols for the in vitro generation of astrocytes with defined maturity states and phenotypic properties [39].

### 4.1. In Vitro Model Characterization

In this context, our in vitro NSP differentiation mouse model enabled us to conduct kinetic analyses along the process of astrocytic differentiation. In this model, under specific culture conditions, NSCs retain their multipotential properties, thus being able to give rise to neurons, astrocytes, and oligodendrocytes. For our experiments, we differentiated NSPs into astrocytic populations by cultivating them in the presence of serum, as previously described [21]. In these conditions, we detected the highest *Gfap* expression levels at 48 and 72 h, whereas its expression decreased after 1 week. This *Gfap* expression pattern is supported by previous in vivo studies, documenting that although GFAP is a reliable marker of reactive astrocytes, it is only variably detectable in mature astrocytes of the healthy mouse central nervous system [40]. Intriguingly, although GFAP is a reliable marker for reactive astrocytes, within our model we investigated the effect of inflammation in a different context than in mature astrocytes, as we treated NSCs with TNF and analyzed the resulting astrocytic phenotype. From our results, it appears that TNF dampens the expression of GFAP in differentiating astrocytes. This observation might be explained by various molecular insights as described below.

### 4.2. Targeted Gene Expression and Genome-Wide Transcriptional Analyses

We have previously shown that TNF is a key regulator of astrocytic reprogramming into neural progenitor-like cells [41]. In addition, TNF treatment of in vitro differentiated murine astrocytes resulted in modifications of their histone chromatin profile at the promoters of genes related to cell cycle, stemness or neuronal fate, thus indicating that inflammatory events intervene in such regulatory pathways [27]. In the present study, RNA-seq analyses showed major gene expression changes along the physiological process of astrocytic differentiation, which is modulated by TNF, suggesting the establishment of discrete astrocytic populations between normal and inflammatory conditions. More specifically, TNF-treated NSC-derived astrocytes exhibited significantly increased expression levels of classical reactive astrocytic marker genes, including *Timp1*, *Lcn2*, *Icam1,* and *Ptx3*, but not *Gfap*, which is the classic indicator of astrogliogenesis. In line with these observations, reactive astrocytes in inflammatory environments display increased expression levels of various proteins that are absent or weakly expressed in resting states. Among those proteins, LCN2 and TIMP1 are highly up-regulated in reactive astrocytes, with LCN2 considered as a marker of astrogliosis [28]. ICAM-1 is aberrantly expressed by astrocytes in CNS pathologies, such as multiple sclerosis, experimental allergic encephalomyelitis and Alzheimer’s disease, suggesting a possible role for ICAM-1 in these disorders [42]. Lastly, *Ptx3* is considered as a marker of A2 reactive astrocytes [29,43]. Hence, TNF-treated NSC-derived astrocytes express markers of both A1 and A2 reactive types, either representing a distinct type of astrocytes that do not exclusively relate to the described dichotomous distribution or signifying a hybrid population.

Notably, the loss of normal astrocytic functions and not only reactivity contributes to the early pathologic events of neurological diseases, as described for example in multiple sclerosis [44,45]. In this context, we sought to examine genes associated with important astrocytic functions, including glucose import from blood, the primary energy source for generating ATP in the brain, and the metabolism of glycogen, which represents an important energy reserve synthesized from glucose in astrocytes [46]. Indeed, astrocytes represent an exclusive reserve of glycogen in the brain to produce lactate as an energy source transported to active neurons [47,48,49,50]. Thus, the detection of decreased expression levels of genes associated with glucose and glycogen metabolism under TNF exposure suggests an impairment of key functional activities when NSCs differentiate into astrocytes under inflammatory conditions. Of note, we have previously demonstrated that TNF treatment of primary mature astrocytes induces the decrease of genes related to glycogen metabolism. Specifically, the expression levels of glycogen phosphorylase and protein targeting to glycogen were strongly down-regulated following TNF exposure [41].

### 4.3. Genome-Wide Chromatin Accessibility Profiling

The ATAC-seq method enabled us to analyze the accessible chromatin landscape of differentiating astrocytes and the extent of its modifications upon inflammation, while providing insights into gene transcription complexity. We observed a general decrease of chromatin accessibility along NSC differentiation into astrocytes, thus suggesting a more lineage-restricted profile. However, upon TNF treatment, chromatin accessibility at the promoter region of specific NSC, astrocytic, and inflammatory markers was up-regulated, indicating that inflammation-induced astrocytes may retain some stemness vestige. The next step would be to identify which epigenetic mechanisms, such as DNA methylation or histone modifications, may be linked to the transcriptional and chromatin accessibility alterations that we identified either in the physiologic differentiating astrocytes or under inflammatory conditions. Intriguingly, a recent study showed that aberrant transcription in gliomas does not result from DNA methylation mechanisms, but rather from DNA methylation-independent modifications [51].

### 4.4. Footprint Analyses

Footprint analyses identified, among several modulated TF binding motifs across accessible chromatin regions, decreased levels of the downstream effector of the Notch pathway, HES1, when comparing differentiated astrocytes at 72 h with NSCs. Since HES1 is involved in the repression of pro-neural gene expression and maintenance of NSCs [52,53,54], decreased levels of its binding motifs at later time points are in line with a lineage-restricted fate of differentiated astrocytes when compared to NSCs. Further, we detected increased levels of NRF2 binding motifs across accessible chromatin regions at 24 h under inflammatory conditions. NRF2 is a master regulator of stress-induced antioxidant response inducing the expression of antioxidants and cytoprotective genes [55]. In the brain, NRF2 is involved in astrocytic neuroprotective pathways via their interactions with neurons [55,56]. Interestingly, in our dataset, the NRF2 target gene *Cebpb* [57] was both upregulated and accessible in the presence of TNF. Taken together, our results point towards NRF2 as a key TF counterbalancing inflammation and triggering self-protection in differentiating astrocytes exposed to TNF.

### 4.5. RNA-seq & ATAC-seq Correlation Analyses

Our correlation analyses are consistent with previous correlation studies addressing gene expression and chromatin accessibility, thus providing further support to our data [58]. Notably, correlation levels were enhanced when looking at NSC, astrocytic, and inflammatory markers. Further, our chromatin accessibility and gene expression analyses resulted in similar biological processes and pathways, thus indicating concordant modules of regulation at these molecular levels.

### 4.6. Reconstructed Gene Regulatory Networks

GRNs inferred from our transcriptional analyses enabled the identification of TFs underlying the astrocytic differentiation under physiological and inflammatory conditions. Different pathways are known to be sequentially involved in the astrocytic differentiation process, with the Notch, STAT3, and BMP signaling pathways being the most important ones implicated in this process [59]. At 24 h of differentiation under normal conditions, the RNA of several TFs related to these pathways were highly represented in the GRN. These results validate once again the relevance of our in vitro model, which is often described in the literature as very different from the differentiation processes obtained in vivo [60]. Indeed, under our established differentiation conditions in the absence of growth factors and mitogen cocktails found in other studies [60,61], we obtained an efficient and reproducible process of astrocytic differentiation. Among the key detected regulators, *Esr1*, *Nr3c1*, *Foxp1*, *Runx2,* and *Nfia* appear particularly interesting. For example, *Esr1*-dependent signaling in astrocytes is involved in the regulation of several important pathways, such as PI3K-AKT, ERK, and JAK-STAT, and plays a crucial role in their metabolic control of providing nutrients to the neurons [62,63]. *Nr3c1*, also known as glucocorticoid receptor, is an important component of the metabolic functions of astrocytes and is required for memory formation [64]. NSCs express *Foxp1* promoting their maintenance and renewal [65], and is required for differentiation towards astrocytes [66]. *Runx2* promotes maturation-associated transcriptional changes in astrocytic cell culture models [61]. Lastly, *Nfia* contributes to the gliogenic switch enabling rapid derivation of functional human astrocytes from pluripotent stem cells [67]. Its expression is required for the maintenance of the astrocytic morphological complexity and its deficiency compromises the astrocyte–neuron communication [68]. Notably, most of the detected TFs show reduced chromatin accessibility after 24 h of differentiation, thus suggesting that they are implicated in the first step of differentiation, but not in the astrocytic maturation. The TFs identified at 72 h relate to the maturation and development of astrocytic functions. Compared to earlier time points, the network is less complex, characterized by the reduction of the number of TFs composing the GRN with a robust representation of TFs related to retinoic acid (RA). In astrocytes, GFAP expression induced by RA is mediated by the activation of RA receptor alpha (Rara), which downstream activates the PI3K pathway and binds to the STAT3-p300/CBP-SMAD complex [69]. The PI3K-AKT pathway also plays crucial roles in plasmalemmal glutamate transport [70] and synthesis of neuroprotective RANTES in astrocytes [71]. The presence of *Sox2* in the network reflects a certain degree of immaturity still at 72 h of differentiation. In fact, SOX2 acts upstream the Notch pathway to maintain cell proliferation potential, and its expression in the brain is restricted to neural stem and progenitor cells, glial precursors and proliferating astrocytes [72]. This observation is corroborated by the presence in the GRN of the effector of the Notch pathway, *Rbpj*, whose reduced accessibility suggests that the Notch pathway has lost part of its activity. Interestingly, reduction of Notch signaling in astrocytes promotes neurogenesis and absence of the Notch coactivator RBPJ activates neurogenesis in astrocytes [73]. Lastly, the TGFB1-SMAD signaling, with *Smad3* being among the TFs whose accessibility is increased, is involved in the fate commitment process of astrocytes [74].

Under inflammation, the TFs that are involved in the GRN at 24 h represent both astrocytic development and inflammatory responses. This is reflected by the overlap of regulatory hubs from the normal condition and TFs representing the inflammatory part of the process (e.g., *Stat3*, *Nfkb2*, *Runx1*, *Atf3*, *Relb*). STAT3 and NF-κB regulate the function and development of cells under inflammation [75,76,77]. The roles of NF-κB and STAT3 in colon, gastric, and liver cancers have been extensively investigated, and the activation and interaction between STAT3 and NF-κB plays a fundamental role in the control of the communication between cancer cells and inflammatory cells [78,79]. Specifically, STAT3 plays an important role in the development of astrocytes, as it binds to the promoter region of the *GFAP* gene inducing its expression, and its absence results in defects during astrogliogenesis [80]. The decrease of chromatin accessibility at the promoter region of *Stat3* may reflect the partial inhibition of the astrocytic differentiation induced by NF-κB activation. NFKB1 and NFKB2 are critical effectors of the canonical NF-κB pathway regulating expression of genes involved in inflammatory and cell-survival responses [81]. Further, *Runx1* codes for an important TF involved in the astrocytic development, and its overexpression is linked to differentiation along astrocytic and neuronal lineages [82]. Lastly, *Cebpb* is involved in the regulation of multiple pathways associated with inflammatory responses in astrocytes [83]. *Atf3* expression in astrocytes is induced by NRF2, a key regulator of stress-induced antioxidant response [84]. It is interesting to note that several TFs, such as *Esr1*, are present in the GRN both under normal and inflammatory conditions at 24 h of differentiation, thus suggesting that although inflammation affects the astrocytic differentiation, patterns of the normal process still drive the process of differentiation.

At 72 h under inflammation, the GRN is composed of TFs involved in multiple metabolic functions. For example, *Rarg*, *Rorc,* and *Rxra* are involved in the RA pathway, while *Thrb* is the thyroid hormone receptor. Astrocytes are the mediators of thyroid metabolism, which is linked to the proliferation and differentiation of astrocytes and also promotes neuronal development via astrocytes [85]. The corresponding GO terms support these processes related to metabolism (e.g., “response to steroid hormone”, “response to corticosteroid”, and “response to glucocorticoid”). Overall, the identified TFs support the tight interplay between development and inflammation pathways influencing astrocytic functional activities.

## 5. Conclusions

Taken together, our results show that an inflammatory environment affects NSC differentiation into astrocytes. Indeed, we here demonstrate that exposure of NSCs to TNF, a master pro-inflammatory cytokine, skews serum-differentiated astrocytes towards a specific phenotype characterized by impaired key functional activities ascribed to normal astrocytes, including glucose and glycogen metabolism. These divergent astrocytes may inadequately support the underlying neuronal network, would it be in its nascent phases, e.g., by affecting synaptogenesis; or in the adult brain, e.g., by inadequately providing nutrients to neurons, thus modifying the fine-tuned neuronal activities. Overall, these changes might contribute to the development and progression of specific neurological disorders, including neurodevelopmental diseases and brain tumors.

## Figures and Tables

**Figure 1 cells-12-00948-f001:**
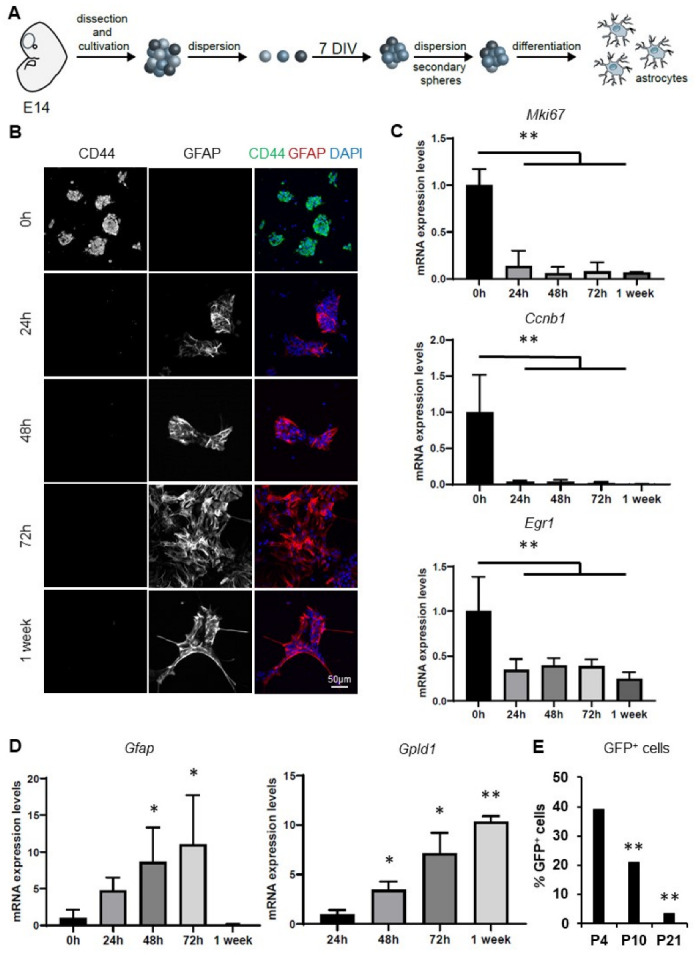
In vitro differentiation of neurospheres (NSPs) into astrocytes. (**A**) Schematic showing the in vitro cellular model of primary neurospheres (NSPs) derived from NSCs isolated from the ventricular zone at embryonic day-14 (E14) of the mouse embryos and the subsequent passages to differentiate them into astrocytes. (**B**) Immunostaining analyses of CD44 (green) and GFAP (red) of NSPs and differentiating astrocytes obtained from primary murine cultures of NSPs cultivated under proliferation conditions (DMEM F12, 1% B27 without vitamin A, antibiotics and 20 ng/mL EGF) or differentiated into astrocytes (DMEM, 10% FBS and antibiotics) for 24 h, 48 h, 72 h, and 1 week. Nuclei were counterstained with DAPI (in blue). Scale bar: 50 μm. (**C**,**D**) RT-qPCR analyses showing mRNA expression levels of (**C**) the cell cycle genes *Mki67* and *Ccnb1*, the NSC marker *Egr1*, (**D**) the astrocytic marker *Gfap,* and the astrocyte-enriched gene *Gpld1*. Results are expressed as mean ± standard deviation. Each time point is shown relative to a specific condition set to 1. * *p* < 0.05; ** *p* < 0.01, *n* ≥ 4. (**E**) Quantification of GFP^+^ cells from mouse forebrain cell suspensions of *hGFAP::*eGFP transgenic mice by FACS at postnatal day 4 (P4), P10, and P21. Bars represent the mean ± SEM. ** *p* < 0.01, *n* = 3.

**Figure 2 cells-12-00948-f002:**
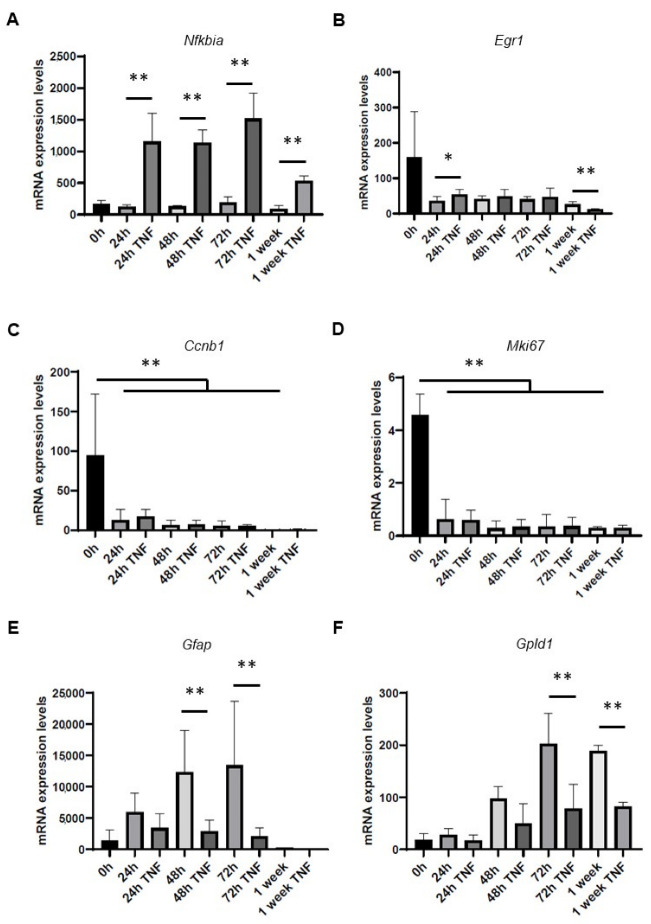
Gene expression levels of NSC and astrocyte markers following treatment of NSCs with TNF during astrocytic differentiation. NSPs were differentiated into astrocytes with or without exposure to TNF (50 ng/mL) for 24 h, 48 h, 72 h, and 1 week. RT-qPCR analyses show the mRNA expression levels of (**A**) *Nfkbia*, (**B**) *Egr1*, (**C**) *Ccnb1*, (**D**) *Mki67*, (**E**) *Gfap,* and (**F**) *Gpld1*. Results are expressed as mean ± standard deviation of 2^−ΔCt^ normalized to beta actin levels. * *p* < 0.05; ** *p* < 0.01, *n* ≥ 4.

**Figure 3 cells-12-00948-f003:**
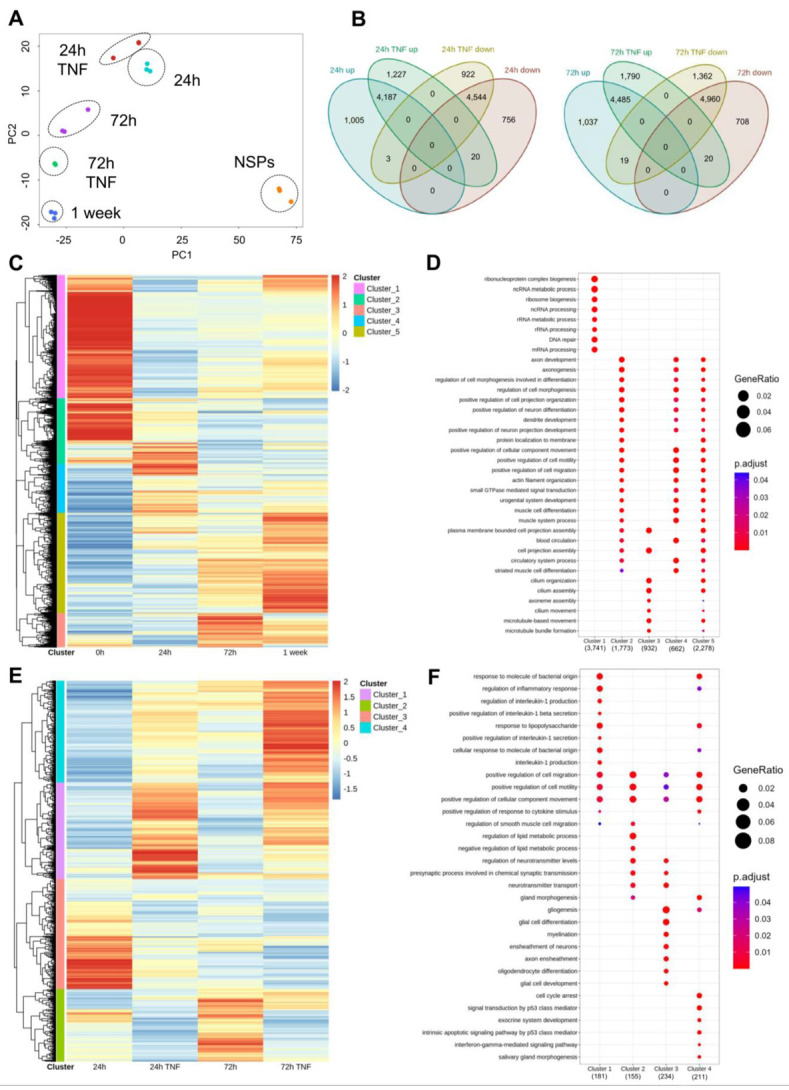
Transcriptomic signature of differentiating astrocytes under normal and inflammatory conditions. (**A**) Representation of first two principal components (PC) analysis of the top 1000 most variable genes across all the conditions analyzed by RNA-seq. (**B**) Venn diagrams showing the number of significantly differentially expressed genes (*p* < 0.05) and their overlap across comparisons 24 h TNF vs. 24 h (left) and 72 h TNF vs. 72 h (right). (**C**) Heatmap showing normalized (z scores) raw gene expression counts of differentially expressed genes under normal conditions based on absolute log2 fold change from comparisons between the different time points (1 week vs. 72 h, 72 h vs. 24 h, and 24 h vs. 0 h). (**D**) Dot plot showing gene set enrichment analysis of genes in clusters identified in the heatmap (Figure 3C). (**E**) Heatmap showing normalized (z scores) raw gene expression counts under inflammatory conditions (24 h TNF vs. 24 h and 72 h TNF vs. 72 h) based on absolute log2 fold change. (**F**) Dot plot showing gene set enrichment analysis of genes in clusters identified in the heatmap (Figure 3E).

**Figure 4 cells-12-00948-f004:**
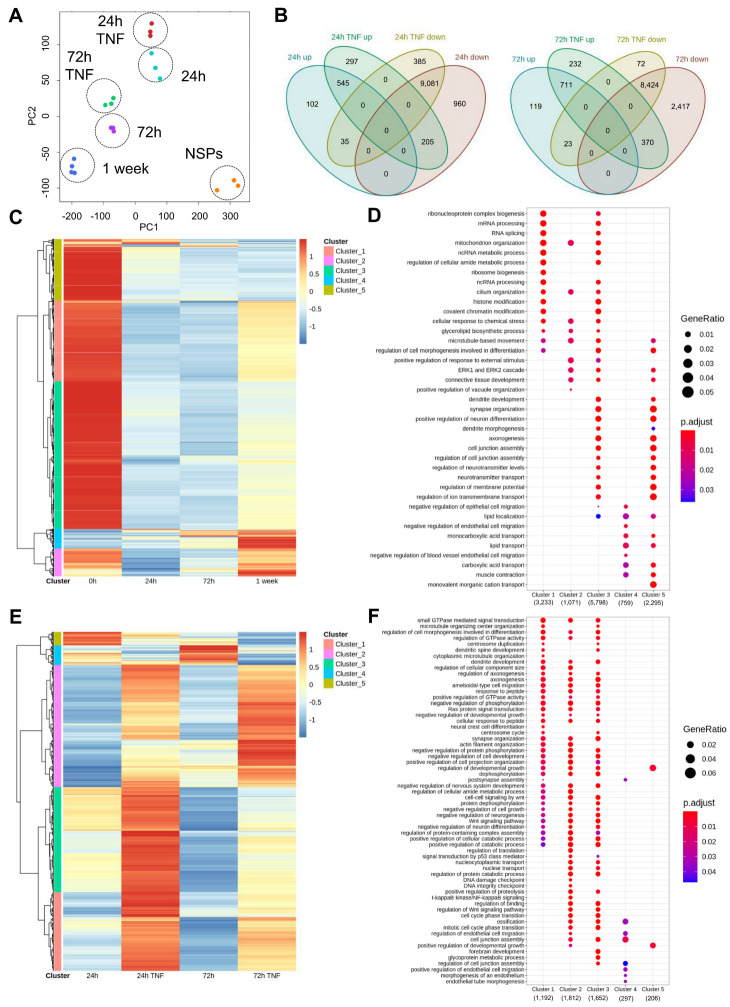
Chromatin accessibility profile of differentiating astrocytes under normal and inflammatory conditions. (**A**) Representation of first two principal components (PC) analysis of the top 1000 most variable genes across all the conditions analyzed by ATAC-seq. (**B**) Venn diagrams showing the number of significantly differentially accessible peaks (*p* < 0.05) and their overlap across comparisons 24 h TNF vs. 24 h (left) and 72 h TNF vs. 72 h (right). (**C**) Heatmap showing normalized (z scores) raw expression of differential peaks under normal conditions based on absolute log2 fold change from comparisons between the different time points (1 week vs. 72 h, 72 h vs. 24 h, and 24 h vs. 0 h). (**D**) Dot plot showing gene set enrichment analysis of genes in clusters identified in the heatmap (Figure 4C). (**E**) Heatmap showing normalized (z scores) raw expression peak counts under inflammatory conditions (2 h TNF vs. 24 h and 72 h TNF vs. 72 h) based on absolute log2 fold change. (**F**) Dot plot showing gene set enrichment analysis of genes in clusters identified in the heatmap (Figure 4E).

**Figure 5 cells-12-00948-f005:**
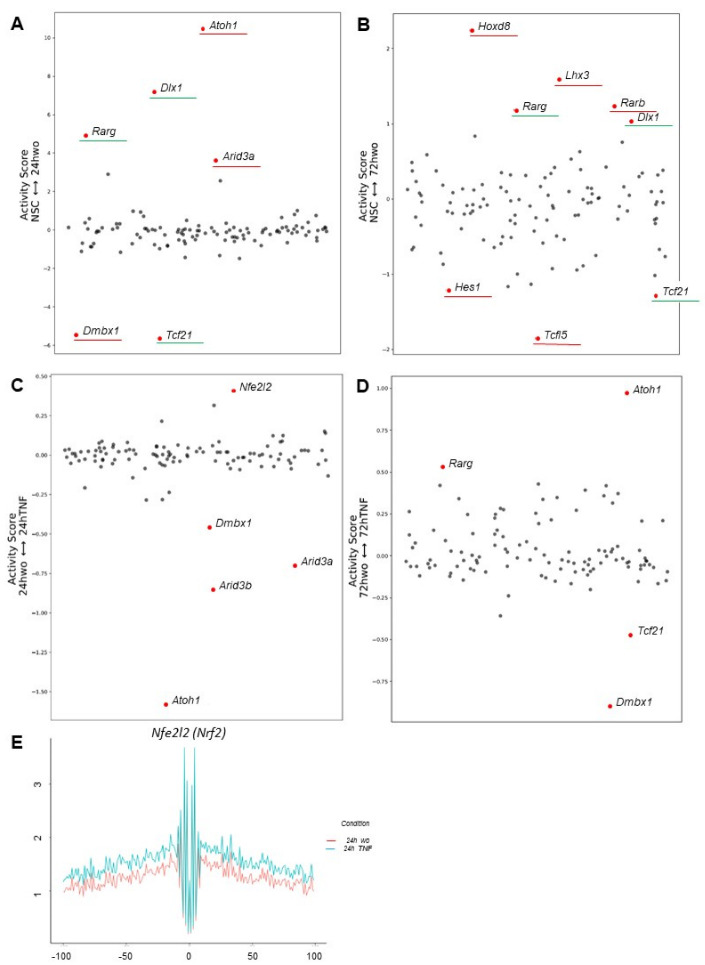
Chromatin footprinting activity under normal and inflammatory conditions. (**A**–**D**) Activity score plots at 24 and 72 h time points under (**A**,**B**) normal and (**C**,**D**) inflammatory conditions. In (**A**,**B**), shared TF binding motifs at 24 and 72 h are underlined in green, while unique TF binding motifs are in red. (**E**) Comparison of the activity of *Nfe2l2 (Nrf2)* gene based on motif enrichment analysis at 24 h between normal (red line) and inflammatory (blue line) conditions.

**Figure 6 cells-12-00948-f006:**
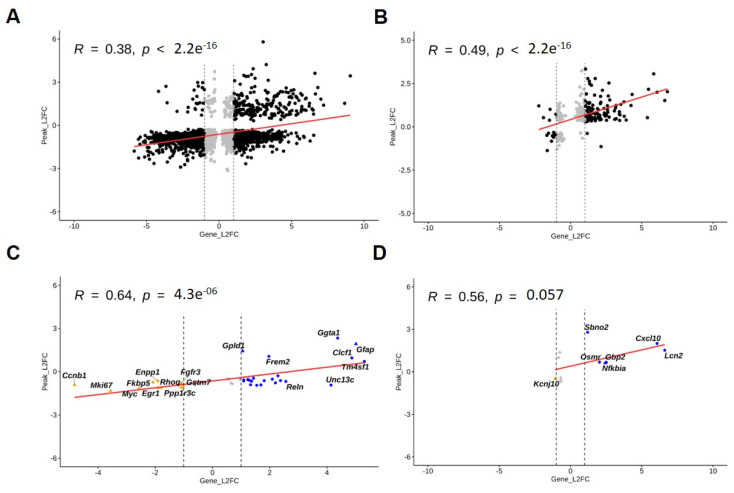
Correlation analyses of transcriptional and chromatin accessibility changes upon the differentiation process under normal and inflammatory conditions. Scatter plots for combined RNA-seq and ATAC-seq data (**A**,**B**) at the genome-wide level and (**C**,**D**) at specific NSC, resting and reactive astrocytic gene loci at 24 h under (**A**,**C**) normal and (**B**,**D**) inflammatory conditions. Plots display Pearson’s correlation coefficient indicating correlation levels between RNA-seq and ATAC-seq data, along with corresponding *p*-values (considering differentially expressed genes with adjusted *p* value < 0.05 and differentially accessible peaks with false discovery rate (FDR) <0.05). Blue and orange dots represent marker genes, which are significantly differentially expressed (adjusted *p* value < 0.05), have more than 2-fold positive change (upregulation) or more than 2-fold negative change (downregulation), respectively, in gene expression and have a significantly differentially accessible peak (FDR < 0.05) in their promoter region. Red lines represent the regression line of correlation between fold change in gene expression and accessibility.

**Figure 7 cells-12-00948-f007:**
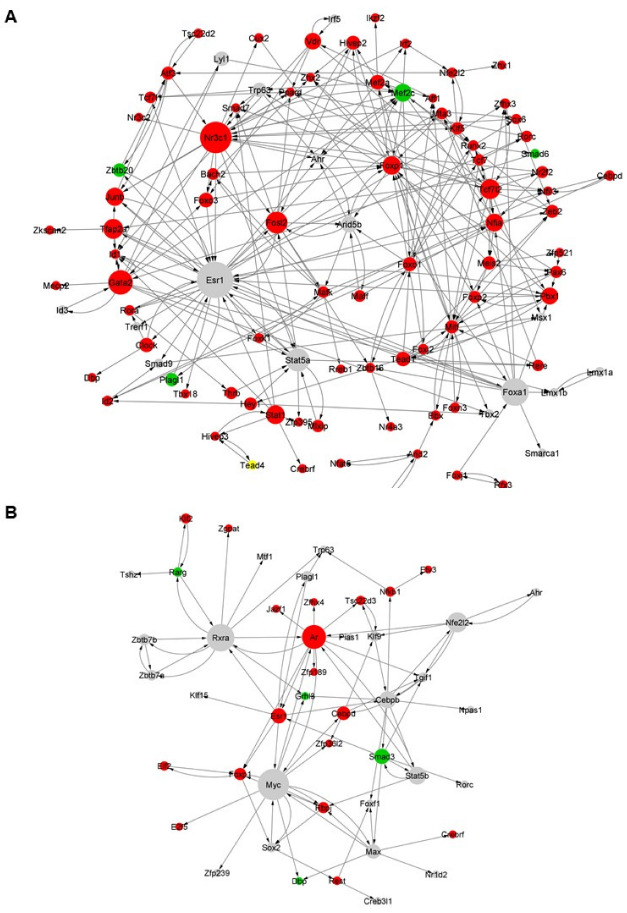
Reconstructed gene regulatory networks under normal conditions. TF-TF regulatory network at (**A**) 24-h and (**B**) 72-h time points. Color legend represents differential accessibility of the TFs (green: upregulated; red: downregulated; grey: not differentially accessible).

**Figure 8 cells-12-00948-f008:**
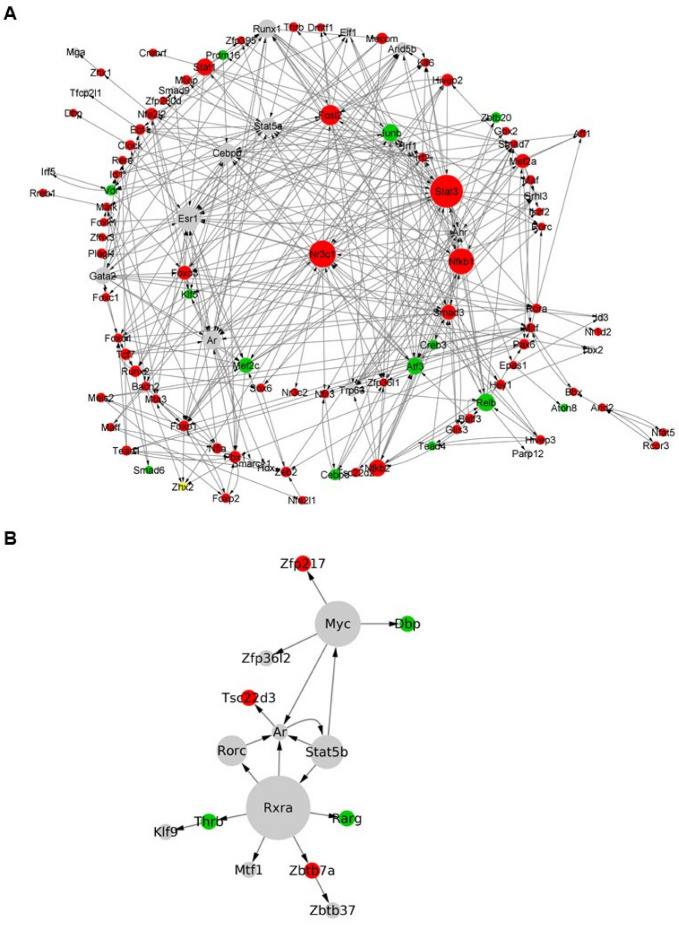
Reconstructed gene regulatory networks under inflammatory conditions. TF-TF regulatory network at (**A**) 24-h and (**B**) 72-h time points. Color legend represents differential accessibility of the TFs (green: upregulated; red: downregulated; grey: not differentially accessible).

## Data Availability

Gene Expression Omnibus (GEO) database—Accession number GSE225729.

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
