# Peer review of "Transcriptional and Chromatin Accessibility Profiling of Neural Stem Cells Differentiating into Astrocytes Reveal Dynamic Signatures Affected under Inflammatory Conditions"

_cells, 2023, doi:10.3390/cells12060948_

Round 1

Reviewer 1 Report

 In the manuscript entitled "Transcriptional and chromatin accessibility profiling of neural stem cells differentiating into astrocytes reveal dynamic signatures affected under inflammatory conditions," Pavlou M. et al. established an in vitro model for the differentiation of NSCs into astrocytes and monitored the development/maturation of these cells over time by analyzing astrocyte-specific markers such as CD44, GFAP, and GLAST at the protein level by immunocytochemistry or at the gene level by analysis of mRNA expression. Gene expression analysis of proliferation (Mki67 and Ccnb1) and stem cell markers (Egr1) and the astrocyte-enriched gene Gpld1 complemented further characterization of astrocytic differentiation. The authors induced inflammatory conditions for developing astrocytes in vitro by exogenous administration of TNF and performed extensive bioinformatics to determine key regulators underlying physiological astrocytic developmental processes and during inflammatory responses. The study appears to have been well-designed and conducted, state-of-the-art methods have been used and the data largely justify the conclusions. Since the development of astrocytes may be the basis for their future heterogeneity and functional diversity, the topic of the study is timely and of interest to the readers of the journal. Some minor issues are listed below that need to be fixed before release.

In the Results section, the mRNA expression levels for GFAP, Gpld1, Ccnb1, and Egr in Fig. 1 and the mRNA expression levels for the same genes and identical experimental groups in Fig. 2 differ by several orders of magnitude. Please explain.

The determination of developmental and maturational markers of astrocytes is critical. In this context, the authors should be cautious in referring to GFAP exclusively as a pan-astrocytic immature marker (Results, Lane 310) or as mainly undetectable in mature astrocytes of the healthy central nervous system (Discussion, Lane 652), because GFAP has been shown to label many healthy mature astrocytes (in the reference cited by authors 10.1007/s00401-009-0619-8, and elsewhere 10.1016/j.pneurobio.2011.01.005). Since GFAP is a reliable marker for reactive astrocytes, how do the authors explain that TNF does not significantly induce the gene for this marker in NSC-derived astrocytes 24 hours after administration and even causes a significant decrease 72 hours after administration (Figure S3-D)?

In the Discussion (Lane 680-683), the authors should mention whether TNF (inflammation) induces changes in glucose and glycogen metabolism in mature astrocytes and comment on their findings in light of existing data.

In the Discussion, the authors refer to NRF2 as a key TF that orchestrates the immune response and as a master regulator of inflammatory responses (Lane 711 and 780). This should be corrected because, according to the available scientific data and authors, NRF2 is the main regulator of the stress-induced antioxidant response by inducing the expression of antioxidant and cytoprotective genes. Therefore, NRF2 should not be considered a master regulator of inflammatory responses, but rather a counterbalance to inflammation and a trigger of self-protection.

At the end of the manuscript, it would be beneficial for the authors to draw a general conclusion and provide a more comprehensive picture of their findings.

Author Response

Reviewer #1

In the manuscript entitled "Transcriptional and chromatin accessibility profiling of neural stem cells differentiating into astrocytes reveal dynamic signatures affected under inflammatory conditions," Pavlou M. et al. established an in vitro model for the differentiation of NSCs into astrocytes and monitored the development/maturation of these cells over time by analyzing astrocyte-specific markers such as CD44, GFAP, and GLAST at the protein level by immunocytochemistry or at the gene level by analysis of mRNA expression. Gene expression analysis of proliferation (Mki67 and Ccnb1) and stem cell markers (Egr1) and the astrocyte-enriched gene Gpld1 complemented further characterization of astrocytic differentiation. The authors induced inflammatory conditions for developing astrocytes in vitro by exogenous administration of TNF and performed extensive bioinformatics to determine key regulators underlying physiological astrocytic developmental processes and during inflammatory responses. The study appears to have been well-designed and conducted, state-of-the-art methods have been used and the data largely justify the conclusions. Since the development of astrocytes may be the basis for their future heterogeneity and functional diversity, the topic of the study is timely and of interest to the readers of the journal. Some minor issues are listed below that need to be fixed before release.

Re: We thank the reviewer for the positive evaluation of our work and for recognizing the relevance of our analyses.

In the Results section, the mRNA expression levels for GFAP, Gpld1, Ccnb1, and Egr in Fig. 1 and the mRNA expression levels for the same genes and identical experimental groups in Fig. 2 differ by several orders of magnitude. Please explain.

Re: We apologize for not having being clear here. For mRNA analyses by qPCR, in Fig. 1 the obtained absolute values were showed compared to a specific time point condition set to 1 (i.e. 0h for Mki67, Ccnb1, Egr1, Gfap and 24h for Gpld1). We added this information in the corresponding legend. In Fig. 2, the absolute values were not normalized towards a specific condition.

The determination of developmental an maturational markers of astrocytes is critical. In this context, the authors should be cautious in referring to GFAP exclusively as a pan-astrocytic immature marker (Results, Lane 310) or as mainly undetectable in mature astrocytes of the healthy central nervous system (Discussion, Lane 652), because GFAP has been shown to label many healthy mature astrocytes (in the reference cited by authors 10.1007/s00401-009-0619-8, and elsewhere 10.1016/j.pneurobio.2011.01.005). Since GFAP is a reliable marker for reactive astrocytes, how do the authors explain that TNF does not significantly induce the gene for this marker in NSC-derived astrocytes 24 hours after administration and even causes a significant decrease 72 hours after administration (Figure S3-D)?

Re: We thank the reviewer for raising this critical point. Although we definitely agree that GFAP is a reliable marker for reactive astrocytes, within our model we investigated the effect of inflammation in a different context than in mature astrocytes, as we treated NSCs with TNF and analysed the resulting astrocytic phenotype. From our results, it appears that TNF dampens the expression of GFAP in differentiating astrocytes. This observation might be explained by various molecular insights. For example, the crosstalk between the NFkB and STAT3 pathways, the latter playing an important role in the development of astrocytes as it binds to the promoter region of GFAP inducing its expression and its absence results in defects during astrogliogenesis [78], might explain these results as mentioned in the Discussion section (Lane 767-778).

In the Discussion (Lane 680-683), the authors should mention whether TNF (inflammation) induces changes in glucose and glycogen metabolism in mature astrocytes and comment on their findings in light of existing data.

Re: According to the reviewer’s suggestion, we added the following sentence (Lane 684-687): “Of note, we have previously demonstrated that TNF treatment of primary mature astrocytes induces the decrease of genes related to glycogen metabolism. Specifically, the expression levels of glycogen phosphorylase (PYGL) and protein targeting to glycogen (PTG) were strongly down regulated following TNF exposure [39]”.  

In the Discussion, the authors refer to NRF2 as a key TF that orchestrates the immune response and as a master regulator of inflammatory responses (Lane 711 and 780). This should be corrected because, according to the available scientific data and authors, NRF2 is the main regulator of the stress-induced antioxidant response by inducing the expression of antioxidant and cytoprotective genes. Therefore, NRF2 should not be considered a master regulator of inflammatory responses, but rather a counterbalance to inflammation and a trigger of self-protection.

Re: We thank the reviewer for highlighting this inaccuracy. We stated the role of NRF2 in counterbalancing inflammation and triggering self-protection as well as a key regulator of stress induced antioxidant response instead of acting as a master regulator of inflammatory responses in the revised version of the manuscript (Lane 715 and 784). 

At the end of the manuscript, it would be beneficial for the authors to draw a general conclusion and provide a more comprehensive picture of their findings.

Re: We thank the reviewer for this suggestion. We have accordingly added a Conclusion section explaining the overall significance of our findings (Lane 799-809).

Reviewer 2 Report

The ms. of Maria Angeliki S. Pavlou et al Transcriptional and chromatin accessibility profiling of neural 2 stem cells differentiating into astrocytes reveal dynamic signa-3 tures affected under inflammatory conditions is a well written paper that deals on the effect that inflammatory or pathologic environments interplay in the development of astrocytes properties such are transcriptional and chromatin accessibility profiling of neural 2 stem cells differentiating into astrocytes reveal dynamic signatures affected under inflammatory conditions. Overall the introduction is wells developed, methodology followed is coherent with the experiments designed. Figures and S Figures are clear and illustrative of the data obtained in the study. The discussion is well conducted and finally the cellular experimental model used by authors will be a good test bench form other inflammatory or dysregulated environment far from TNF.

In my opinion the paper is suitable for publication in Cells by solving the minor concern that follows:

Lines 633-635.- These sentence should be explained “Perturbations during astrocytic development may result in neurological diseases, including neurodevelopmental disorders, such as Rett and fragile X syndromes [33] or Down syndrome and autism spectrum disorders [34]” Why astrocytes would constitute the pathobiological basis for these neuronal diseases. Perhaps, neuronal primary lesion induces astrocytic changes? Please clarify this point.

Author Response

Reviewer #2

The ms. of Maria Angeliki S. Pavlou et al Transcriptional and chromatin accessibility profiling of neural 2 stem cells differentiating into astrocytes reveal dynamic signa-3 tures affected under inflammatory conditions is a well written paper that deals on the effect that inflammatory or pathologic environments interplay in the development of astrocytes properties such are transcriptional and chromatin accessibility profiling of neural 2 stem cells differentiating into astrocytes reveal dynamic signatures affected under inflammatory conditions. Overall the introduction is wells developed, methodology followed is coherent with the experiments designed. Figures and S Figures are clear and illustrative of the data obtained in the study. The discussion is well conducted and finally the cellular experimental model used by authors will be a good test bench form other inflammatory or dysregulated environment far from TNF.

Re: We thank the reviewer for the nice evaluation of our work and for appreciating the relevance of our cellular experimental model for follow-up studies.

In my opinion the paper is suitable for publication in Cells by solving the minor concern that follows:

Lines 633-635.- These sentence should be explained “Perturbations during astrocytic development may result in neurological diseases, including neurodevelopmental disorders, such as Rett and fragile X syndromes [33] or Down syndrome and autism spectrum disorders [34]” Why astrocytes would constitute the pathobiological basis for these neuronal diseases. Perhaps, neuronal primary lesion induces astrocytic changes? Please clarify this point.

Re: We thank the reviewer to point out this sentence and apologize for not having being clear here. As included in the Conclusion section of the revised manuscript, “these divergent astrocytes may aberrantly support the underlying neuronal network, would it be in its nascent phases, e.g. by affecting synaptogenesis, or in the adult brain, e.g. by abnormally providing nutrients to neurons, thus modifying the fine-tuned neuronal activities. Overall, these changes might contribute to the development and progression of specific neurological disorders, including neurodevelopmental diseases and brain tumors” (Lane 804-809). Specifically for the mentioned diseases characterized by extensive neuronal death, the reviewer’s hypothesis that primary neuronal lesions might induce aberrant astrocytic phenotypes generating a detrimental neuron-astrocyte crosstalk would also be a plausible explanation.
